# Circulation of Respiratory Viruses in Hospitalized Adults before and during the COVID-19 Pandemic in Brescia, Italy: A Retrospective Study

**DOI:** 10.3390/ijerph18189525

**Published:** 2021-09-09

**Authors:** Maria Antonia De Francesco, Caterina Pollara, Franco Gargiulo, Mauro Giacomelli, Arnaldo Caruso

**Affiliations:** 1Institute of Microbiology, Department of Molecular and Translational Medicine, University of Brescia, 25123 Brescia, Italy; arnaldo.caruso@unibs.it; 2Laboratory of Microbiology and Virology, ASST-Spedali Civili, 25123 Brescia, Italy; caterina.pollara@asst-spedalicivili.it (C.P.); franco.gargiulo@asst-spedalicivili.it (F.G.); giacomellimauro@libero.it (M.G.)

**Keywords:** circulation, seasonality, prevention, pandemic, respiratory viruses

## Abstract

Different preventive public health measures were adopted globally to limit the spread of SARS-CoV-2, such as hand hygiene and the use of masks, travel restrictions, social distance actions such as the closure of schools and workplaces, case and contact tracing, quarantine and lockdown. These measures, in particular physical distancing and the use of masks, might have contributed to containing the spread of other respiratory viruses that occurs principally by contact and droplet routes. The aim of this study was to evaluate the prevalence of different respiratory viruses (influenza viruses A and B, respiratory syncytial virus, parainfluenza viruses 1, 2, 3 and 4, rhinovirus, adenovirus, metapneumovirus and human coronaviruses) after one year of the pandemic. Furthermore, another aim was to evaluate the possible impact of these non-pharmaceutical measures on the circulation of seasonal respiratory viruses. This single center study was conducted between January 2017–February 2020 (pre-pandemic period) and March 2020–May 2021 (pandemic period). All adults >18 years with respiratory symptoms and tested for respiratory pathogens were included in the study. Nucleic acid detection of all respiratory viruses was performed by multiplex real time PCR. Our results show that the test positivity for influenza A and B, metapneumovirus, parainfluenza virus, respiratory syncytial virus and human coronaviruses decreased with statistical significance during the pandemic. Contrary to this, for adenovirus the decrease was not statistically significant. Conversely, a statistically significant increase was detected for rhinovirus. Coinfections between different respiratory viruses were observed during the pre-pandemic period, while the only coinfection detected during pandemic was between SARS-CoV-2 and rhinovirus. To understand how the preventive strategies against SARS-CoV-2 might alter the transmission dynamics and epidemic patterns of respiratory viruses is fundamental to guide future preventive recommendations.

## 1. Introduction

Different respiratory viruses, including influenza viruses (FLU), rhinovirus (RV), respiratory syncytial virus (RSV), human metapneumovirus (hMPV), human coronaviruses (CoV), adenoviruses (AdV), and parainfluenza viruses (PIV) contribute to significant morbidity [1] and mortality [2] in adult persons, especially in older adults and in those with underlying comorbidities [3,4,5,6,7,8]. Furthermore, they are responsible for massive economic costs annually worldwide [9].

Although FLU viruses, RSV and common coronaviruses had a seasonal pattern with a peak in the winter months, and RV circulates year-round with a peak incidence in spring and fall, all the other respiratory viruses circulate throughout the entire year. No preventive measures, except for the influenza vaccination, are undertaken to limit the occurrence of these viruses that are accepted as inevitable.

The appearance of severe acute respiratory syndrome coronavirus 2 (SARS-CoV-2) in China in December 2019, and its sudden pandemic spread forced the governments to adopt different non-pharmacological public health preventive measures, such as wearing masks, social distancing, and hand hygiene, to mitigate the diffusion of SARS-CoV-2.

Furthermore, to contain COVID-19 infections, the Italian government used a system based on risk color [10,11], which was updated weekly by using a combination of different criteria among which were hospitalization numbers and the new positive cases numbers.

Following this criterion, the 20 Italian regions may exhibit a very low risk (blank), a low risk (yellow), medium risk (orange) and high risk (red). On the basis of the color risk stratification, different commercial activities might be closed and some activities might be prohibited.

These preventive strategies, in particular physical distancing and the use of masks, may have contributed to reducing the circulation of other respiratory viruses besides SARS-CoV-2 [12,13]. Different studies show that social restrictions had an impact on the spread of both seasonal influenza [14,15,16,17,18] and other respiratory viruses [19,20,21,22].

Only limited data are available in Italy about the prevalence of respiratory viruses during SARS-CoV-2 pandemic [23,24]. Therefore, the aims of this study were:(a)To evaluate the prevalence of different respiratory viruses (FLU A and FLU B, RV, RSV, hMPV, AdV, PIV and human CoVs) during the COVID-19 pandemic period, in samples collected from hospitalized adults, compared to that observed in the three years before the pandemic period;(b)To correlate the possible impact of non-pharmaceutical measures, recommended in response to the COVID-19 pandemic, on the circulation of seasonal respiratory viruses.

## 2. Materials and Methods

### 2.1. Study Design and Patients

This retrospective study was conducted at the Laboratory of Microbiology and Virology, Spedali Civili Hospital, Brescia, Italy. This is a 1650-bed tertiary hospital and is one of the biggest medical centers in Lombardy, in northern Italy. Inclusion criteria were: hospitalized patients aged ≥18 years, and the presence of one or more respiratory symptoms such as shortness of breath, sore throat, cough and fever ≥37.5 °C. We collected data on respiratory virus and/or SARS-CoV-2 testing performed from January 2017 to May 2021 defining two periods as pre-pandemic (January 2017–February 2020) and during pandemic (March 2020–May 2021). The first positive SARS-CoV-2 test and first COVID-19 case admitted to Spedali Civili Hospital occurred on 24 February 2020. Samples collected included nasopharyngeal swabs, bronchoalveolar lavage (BAL) and bronchial aspirate (BAS). Swabs samples were stored at 4 °C until processing.

### 2.2. Detection of Respiratory Viruses and SARS-CoV-2

Total nucleic acid extraction for SARS-CoV-2 was performed by using Seegene Nimbus (Seegene Inc., distributed by Arrow Diagnostics, Genoa, Italy), while total nucleic acid extraction for respiratory viruses was performed by using NucliSENSE^®^ EasyMAG^®^ (BioMérieux Italia, Florence, Italy). Detection of SARS-CoV-2 was performed using a real time reverse polymerase chain reaction (Allplex TM 2019-nCoV assay, Seegene Inc., distributed by Arrow Diagnostics, Genoa, Italy). This single tube assay identified firstly three SARS-CoV-2 gene targets: E, RdRP and N genes, updated then to identify further the S gene.

A multiplex real time PCR was performed to detect all the viruses under investigation in this study (FTD Respiratory pathogens 21, Fast Track Diagnostics, Siemens Healthcare, Milan, Italy).

Upon medical request for specific respiratory pathogens, laboratory diagnosis was performed with the following assays:(a)Respiratory Viral ELITe MGB^®^ Panel (ELITech Italy, Turin, Italy) for influenza viruses (A and B) and RSV detection;(b)FTD HAdV/HMPV/HBoV (Fast Track Diagnostics) for human metapneumovirus A and B and human adenovirus detection;(c)FTD HPIV for PIV (serotypes 1, 2, 3 and 4) detection;(d)FTD (Fast Track Diagnostics) for human endemic coronaviruses (HKU1, NL63, 229 and OC43) detection.

### 2.3. Statistical Analysis

Descriptive statistics were applied to describe patient characteristics. Comparisons between categorical variables were performed by using the Fisher’s exact test with Yates’ correction and the chi-square test as appropriate. Continuous data were analyzed by *t*-test. The significance level was set at 0.05.

## 3. Results

During the study period, 12,483 patients were tested for different respiratory viruses. Of these, 10,121 were analyzed from January 2017 to February 2020 and 2362 from March 2020 to the end of this study (May 2021). Patient characteristics were summarized in Table 1. 

As the number of COVID-19 infections increased from March 2020 (Figure 1), the number of tests required from clinicians during the pandemic for the other respiratory viruses decreased by less than a quarter, compared to the pre-pandemic period (2362 vs. 10,121) (Table 1).

Moreover, the overall test positivity for them was lower during the pandemic than in the previous years (2.7% vs. 14.6% %, *p* < 0.0001) (Table 1). Demographic characteristics show that in the adult population tested during pandemic period, male gender was more prevalent (55.5% vs. 50.3%, *p* < 0.0001) and the median age was lower (63.5 vs. 65.1, *p* < 0.0001) compared to that tested during the pre-pandemic period. No statistical difference for gender was found between subjects positive for respiratory viruses analyzed during the two periods.

Group 45–64 years was more prevalent during the pandemic period compared to that in the pre-pandemic period (39.7% vs. 29.7%, *p* < 0.0001), while the ≥80 years group accounted for 21.6% of adult patients in pre-pandemic period, compared to that present during the pandemic (13.6%, *p* < 0.0001) (Table 1). The same results were obtained when virus respiratory positive subjects were stratified for age (Table 1).

The rate of positivity for each respiratory virus analyzed during the study period is shown in Table 2. The test positivity for FluA, FluB, MPV, PIV, RSV and human CoVs decreased with statistical significance (0.2 vs. 7.6%, *p* < 0.0001; 0.06% vs. 3%, *p* < 0.0001; 0.28% vs. 1.1%, *p* = 0.04; 1.4% vs. 7.3%, *p* < 0.0001 and 0.14% vs. 5%, *p* < 0.0001, respectively). On the contrary, for AdV the decrease was not statistically significant. Conversely, an increase statistically significant of positivity percentage was detected for RV (3.8% vs. 5.7%, *p* = 0.02).

During the seasons analyzed previous the pandemic, we diagnosed 523 influenza A cases and 210 influenza B cases, for an average of 244 combined cases per season. During the pandemic period, only three cases of influenza A and one case of influenza B were detected. The peak for influenza viruses was always reached in the period pre-pandemic in January and February (Figure 1) with a decline of cases in March. However, the decrease in positive tests observed already in late February and March 2020 was maintained in January and February 2021, without the seasonal pattern present in the pre-pandemic period.

Among specimens tested for PIV, hCoVs and MPV, the percentage with a positive result declined in March 2020 and remained suppressed until May 2021, without the seasonal increases that occurred during the pre-pandemic period (Table 2). During the pre-pandemic period, PIV reached a peak during fall (October–November) and spring (March–May) (Figure 1), while during the pandemic period only 1 case of PIV was detected.

The peak circulation of human coronaviruses occurred between January and February during the pre-pandemic period and ranged from 8.1% to 12.5% (Figure 1), disappearing completely during the pandemic period.

The circulation of RSV occurred from winter to spring, reaching a peak between January and February and a positivity rate of 7.3% during the pre-pandemic period, while in the pandemic period it was detected only in March 2020, with a total absence of positive cases in the next months.

Only two cases of human metapneumovirus were detected from March 2020 to May 2021, while during the previous years it circulated mostly from January to April.

From March 2020 to May 2021, adenovirus circulate to lower ranges (0.6%) than those observed during the pre-pandemic period (1.4%).

Contrary to this, the positivity rate of Rhinovirus during the pandemic period increased respect to that observed during the previous years.

In addition, some coinfections were also detected (Table 3). During the pre-pandemic period, RSV was the virus most found in coinfections (7/12, 58.3%). During the pandemic period, the only coinfection observed was between Rhinovirus and SARS-CoV-2 (6 cases).

## 4. Discussion

The non-pharmaceutical interventions, such as social distancing, wearing face masks and increased hand hygiene, used to limit the spread of COVID-19 might also help to prevent other respiratory virus infections, which are transmitted in a similar way to SARS-CoV-2.

According to other studies [16,17,18,19,20,21,22,25,26], our results show that the circulation of respiratory viruses was disrupted during the pandemic, even if the magnitude of this effect was different between the viruses analyzed.

We found that the frequency of influenza and other respiratory viruses (human metapneumovirus; parainfluenza virus 1, 2, 3 and 4; and human coronaviruses) was significantly reduced during the COVID-19 pandemic.

We demonstrated seasonality and sequential seasonal epidemic patterns for these viruses in the previous years analyzed, but with the appearance of SARS-CoV-2, the epidemic patterns of respiratory viruses have undergone a profound change.

After March 2020, influenza virus circulated at very low levels with detection of few cases also in 2021 season. In according with these results, a study from the USA showed a great decrease in the positivity rate during the lockdown and it remained low during the inter-seasonal circulation [27].

The persistence of the effect of the preventive measures against SARS-CoV-2 on respiratory circulation of respiratory viruses is unrecognized. However, the low levels of circulation of influenza viruses during the 2020–2021 seasons might influence the severity of the next influenza season, due to a decreased population immunity owing to the absence of a natural exposure to influenza viruses during these years.

In fact, a study based on mathematical modelling suggests the possibility of a hard rebound, marked by increased morbidity and mortality in the winter of 2021/22 [28], for countries that have experienced years without influenza.

Furthermore, interactions between the SARS-CoV-2 virus and endemic viruses may be very complex. Immunological relationships between viruses, both competitive and cooperative, may have extensive consequences for future infection dynamics [29].

Therefore, with the advent of autumn season, it might be advisable to recommend influenza vaccination to prevent a more widespread disease when influenza virus circulation will restart.

In addition to influenza viruses, parainfluenza viruses, metapneumovirus, human coronaviruses and adenovirus also showed a change in their epidemic patterns, with a marked reduction in their circulation.

Moreover, the positivity rate of respiratory syncytial decreased significantly during the pandemic period. This result was in agreement with other studies that show that positive RSV tests also decreased notably [30,31,32,33], and in disagreement with others that demonstrate a resurgence of RSV circulation [34,35].

Interestingly, a temporal association was observed between the introduction of preventive strategies against COVID-19 (mainly the use of facemasks) and the rate of positive cases for all the above respiratory viruses. All of them had a peak between January and February 2020, but the community mitigation measures were mandatory around mid-March 2020; following that date, we noticed a low circulation of influenza viruses, parainfluenza viruses, metapneumovirus, human coronaviruses, adenovirus and respiratory syncytial virus underlining the possible impact of these preventive strategies on their spread. Our study evidenced that while there was a decline in the positivity rates of other respiratory viruses, the diffusion of rhinovirus seems not to be affected, and it was frequently detected during the pandemic. We observed an increase in its circulation; a finding confirmed by other studies [36,37,38]. The different pattern might be due to their greater stability under environmental conditions (e.g., heat, desiccation, and pH values) than enveloped viruses. Furthermore, a study of Leung et al. [10] showed that facemasks are more effective in filtering out enveloped viruses such as influenza virus and coronavirus than non–enveloped viruses such as rhinovirus.

Rhinovirus is responsible for the common cold, but may cause pulmonary complications in some patients [39]. The signs and symptoms of this infection might be severe and resemble that of COVID-19 in patients with underlying disease [40]. To discriminate between the two viruses is, therefore, important.

Similar results were obtained by a recent study [41], conducted in Shanghai, China on a different study population. Liu et al. found that interventions adopted during the COVID-19 pandemic contributed to the significant reduction of other respiratory viral infections, including RSV, PIV, FluA, FluB, MPV and ADV, except for rhinovirus in a pediatric hospitalized population.

In the pre-pandemic period, we have detected some mixed viral infections where the respiratory syncytial virus was the virus more involved in these coinfections.

Then, during the pandemic, we found only coinfections between SARS-CoV-2 and rhinovirus probably because these are the more frequent circulating viruses. Previous reports have described co-infections between SARS-CoV-2 and common respiratory viruses including rhinovirus, influenza virus, metapneumovirus, parainfluenza virus and respiratory syncytial virus [42,43,44]. Even if it is not clear whether these coinfections are associated with severe disease, determining the coinfection rate and eventual clinical impact on the disease course is important.

This study has some limitations. It was performed at a single center and the reduced hospital accession due to concerns about COVID-19, particularly from persons with milder symptoms might have influenced the number of laboratory confirmed cases for respiratory viruses.

## 5. Conclusions

This study underlines that face coverings, together with social distancing and hand hygiene against COVID-19, are crucial in containing airborne transmitted infections and, therefore, should be implemented during public health emergencies and pandemics or for the future management of respiratory infections. Our data show that these non-pharmaceutical interventions altered the typical seasonal circulation patterns of different respiratory viruses, and that some of them might resurge with the mitigation of these preventive measures, increasing the importance of testing for multiple respiratory pathogens.

## Figures and Tables

**Figure 1 ijerph-18-09525-f001:**
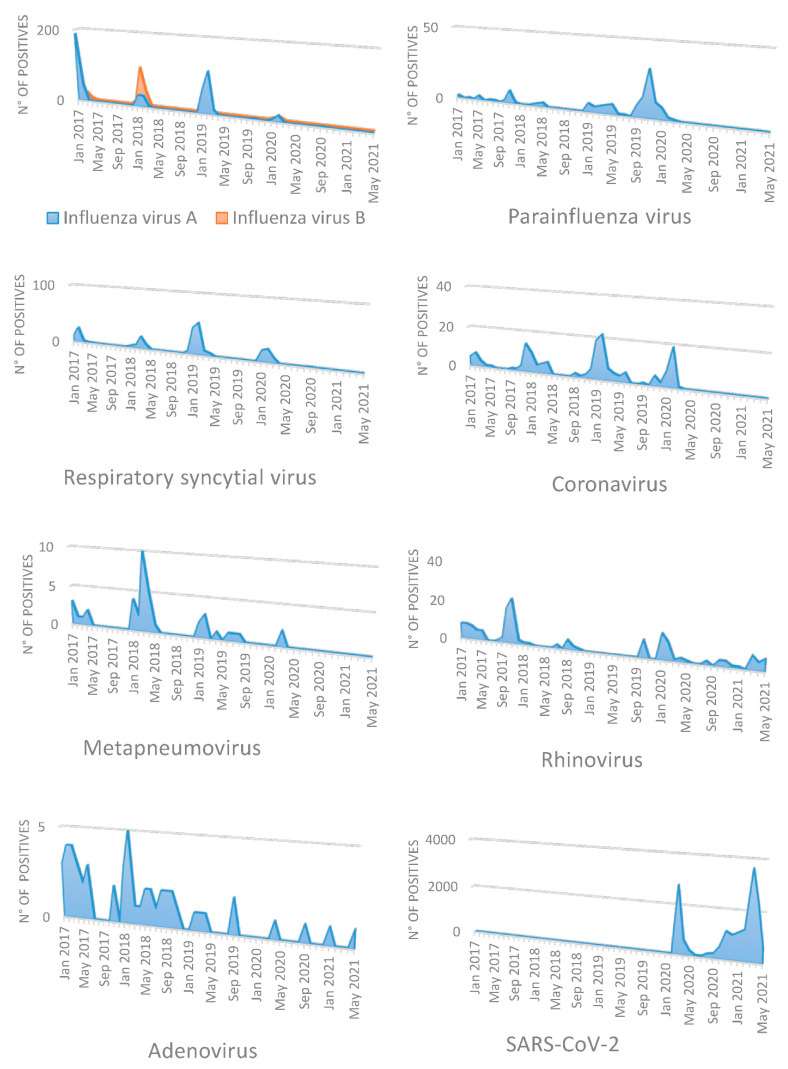
Monthly positive cases of respiratory pathogens in the pandemic period (March 2020–May 2021) in comparison with monthly positive cases in the pre-pandemic period (January 2017–February 2020). Reported positive cases of SARS-CoV-2 are displayed for comparison.

**Table 1 ijerph-18-09525-t001:** Characteristics of patients tested for respiratory viruses other than SARS-CoV-2 during the study period (January 2017–May 2021).

	January 2017–February 2020 (Pre-Pandemic)	March 2020–May 2021 (During Pandemic)	*p*	January 2017–February 2020 (Pre-Pandemic)	March 2020–May 2021 (During Pandemic)	*p*
Samples tested (N°)	10,121	2362				
Positive (N°)	1475	63				
Frequency (%)	14.6	2.7	**<0.0001 ^a^**			
	Total population tested			Positive population		
Demographic characteristics						
Male, *n* (%)	5098 (50.3)	1341 (55.5)	**<0.0001**	878 (59.5)	41 (65)	0.4
Mean Age	65.1	63.5	**<0.0001**	64.02	59.9	0.07
Age in years, *n* (%)						

18–44	1355 (13.2)	261 (11)	**0.002**	240 (16.2)	9 (14.2)	0.86
45–64	3012 (29.7)	940 (39.7)	**<0.0001**	414 (28)	27 (42.8)	**0.01**
65–79	3561 (35.1)	838 (35.4)	0.79	524 (35.5)	24 (38)	0.68
≥80	2193 (21.6)	323 (13.6)	**<0.0001**	297 (20)	3 (4.7)	**0.001**

^a^ Bold character indicates statistical significance.

**Table 2 ijerph-18-09525-t002:** Comparison of positivity rates of respiratory viruses other than SARS-CoV-2 between January 2017–February 2020 (pre-pandemic period) and March 2020–May 2021 (during pandemic period).

	January 2017–February 2020 (Pre-Pandemic)	March 2020–May 2021 (During Pandemic)	*p*
Influenza A virus Test positivity number/total number tests (%)	523/6881 (7.6)	3/1628 (0.18)	**<0.0001 ^a^**
Influenza B virus Test positivity number/total number tests (%)	210/6881 (3)	1/1628 (0.06)	**<0.0001**
Respiratory syncytial virus Test positivity number/total number tests (%)	238/3240 (7.3)	10/734 (1.4)	**<0.0001**
Metapneumovirus Test positivity number/total number tests (%)	38/3240 (1.1)	2/734 (0.27)	**0.04**
Adenovirus Test positivity number/total number tests (%)	48/3240 (1.4)	4/734 (0.6)	0.08
Parainfluenza viruses Test positivity number/total number tests (%)	131/3240 (4)	1/734 (0.14)	**<0.0001**
Rhinovirus Test positivity number/total number tests (%)	123/3240 (3.8)	41/734 (5.6)	**0.02**
Coronaviruses Test positivity number /total number tests (%)	164/3240 (5)	1/734 (0.14)	**<0.0001**

^a^ Bold character indicates statistical significance.

**Table 3 ijerph-18-09525-t003:** Patients with >1 positive respiratory virus detection during the study period (January 2017–May 2021).

	January 2017–February 2020 (Pre-Pandemic)	March 2020–May 2021 (During Pandemic)
Virus detected	No. of patients	
RhV and RSV	1	0
PIV and RSV	1	0
Coronavirus and RSV	2	0
AdV and RSV	1	0
Coronavirus and PIV	1	0
FluA and RSV	1	0
FluA and Coronavirus	2	0
FluA, FluB and Coronavirus	1	0
FluB and RSV	1	0
FluA and PIV	1	0
SARS-CoV-2 and RhV	0	6

Abbreviations: RhV, Rhinovirus; RSV, Respiratory syncytial virus; PIV, Parainfluenza virus, FluA, influenza A virus; FluB, Influenza B virus; AdV, Adenovirus.

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
