# Peer review of "Circulation of Respiratory Viruses in Hospitalized Adults before and during the COVID-19 Pandemic in Brescia, Italy: A Retrospective Study"

_ijerph, 2021, doi:10.3390/ijerph18189525_

Round 1

Reviewer 1 Report

This study evaluated the prevalence of various respiratory viruses during the pre-pandemic period (January 2017-February 2021) and pandemic period (March 2021-May 2021) of COVID-19. As we are still under the pandemic, understanding how the preventive strategies against SARS-CoV-2 alter the transmission of other respiratory viruses is of importance. The study is timely needed. Findings from such a large-sample study (12483 patients) may generate some public health implementation on the current epidemics of COVID-19 and other respiratory diseases.

  1. The authors need to briefly describe the commonly-used preventive measures against COVID-19 and how they can theoretically curb the epidemic of other respiratory diseases.
  2. Typo: “January 2017-February 2021 (pre-pandemic period)” should be “January 2017-February 2020 (pre-pandemic period)” (line 15).
  3. Need a reference for “risk color” so that readers can find detail information about the risk (line 46).
  4. Need to briefly describe the contents of “These preventive strategies”. (line 51).
  5. Please provide more information about the selection of 12483 participants in your study. Was it a convenience sample?
  6. Because the distributions of age and gender differ between the samples in pre- and during the pandemic, analyses should adjust these differences in the comparison of positivity rates between the two periods.
  7. The authors may discuss why the positivity rates of other respiratory virus in the pandemic period were lower than those in the pre-pandemic period. What were the major prevention measures that potentially lead the decline?
  8. What public health implementation of your findings?

Reviewer 2 Report

The study entitled “Circulation of respiratory viruses in hospitalized adults before 2 and during the COVID-19 pandemic: a retrospective study” conducted by M.A. De Francesco et. al is interesting. M.A. De Francesco and co-authors demonstrated the prevalence of diverse respiratory viruses in a period pre-pandemic and during the SARS-CoV-2 pandemic. The study revealed that mostly all respiratory viruses had their circulation diminished in samples analyzed in a hospital in Italy during a SARS-CoV-2 pandemic period. Even though, some minor issues should be addressed in this manuscript before a potential publication. 

1 – Because the authors used samples only from Italy for their study I would recommend changing the title for a more accurate way, specifying that the study was conducted in Italy. References 14 and 20 of the manuscript are good title examples. 

2 – Table 1 shows an import result, where the frequencies of respiratory viruses before and during the pandemic are shown. However, even though the discrepancies between the number of samples of the two groups can be adjusted by statistical methods, it would be interesting to break down the pre-pandemic period in blocks similar to those of the pandemic period time (march 2020 to May 2021 - 14 months) and show that these frequencies are different even when compared side-by-side span of times between the sampling. It could be a supplementary figure that could be appreciated by the readers. and it could make the manuscript even stronger. However, I do not know if it is possible making it.   

3 –Lines 106-107 the authors say “ the number of tests for the other respiratory viruses decreased by less than a quarter compared to January 2017- February 2020,”. I presume the authors wanted to say the number of positive cases for other respiratory viruses decreased by less than a quarter compared to January 2017 – February 2020”. The word test is somewhat confusing in this context. I would suggest to the authors change this word. 

4 – Table 2 has also interesting results, and my comment here is the same as for table 1. 

5 – There is an interesting result in figure 1 that was not addressed by the authors, and it could help to corroborate part of the discussion. It was interesting that IAV and IBV, RSV, Metapneumovirus, Parainfluenza Virus, and Coronavirus had a peak between January/February of 2020. At that time, the use of masks was not mandatory yet, and all of these enveloped viruses were high frequent. After that, with the use of non-pharmaceutical measures (mainly the use of masks) these enveloped viruses were importantly decreased, these finding shown in this figure reinforces in some extent the author's statement (lines 198 to 205) about the role played by masks in avoiding the spread of these viruses.  

6 – On my understanding, because of the lack of references backing the statements in lines 184 to 190 I would consider these somewhat overstatement/speculative. The absence of circulation of the Influenza virus could have a deleterious effect on the population’s immunity in places where the people are not commonly vaccinated. However, IAV vaccination is a measure of diverse countries around the world. Also, viral-viral competition can play a role in this process that is not completely known. I would recommend to the authors add some references to make this statement stronger. 

7 – I agree with the conclusion of the study when it says that the study underlines that non-pharmaceutical interventions against COVID-18 altered typical seasonal circulation patterns […]”. However, as mentioned before, a deeper discussion about this topic should be done to make clearer the author’s aims. In addition, in the last paragraph of the introduction, lines 55 to 59, there is no mention that the authors wanted to evaluate the prevalence of the respiratory viruses in a pre and pandemic period and correlates the findings with a possible role of non-pharmaceutical measures in the findings. Because of this, the conclusion and the aim of the study in the introduction seemed a bit disconnected.    

Reviewer 3 Report

Manuscript ID: ijerph-1348185

Title: Circulation of respiratory viruses in hospitalized adults before and during the COVID-19 pandemic: a retrospective study

Francesco et al has studied the epidemiology of the different respiratory viruses during pre and post COVID-19 pandemic periods in the population of Italy. They reported the decrease in prevalence of other respiratory viruses except rhinovirus during the pandemic period.  However, this study does not reveal novel insight about the circulation pattern of different respiratory viruses with SARS-CoV-2 strain but it might be interesting for local readers and for health authorities in management of the disease. There are few concerns that have been documented here in order to improve this article.

Major Concerns:

  1. Why did this study include only the participants of age group above 18 years and excluded children? Explain.

  1. Similar kind of study is also performed in China (Liu et al. 2021) with similar results (mentioned below). How does this study differ from the study conducted in China regardless of differences in study participants and region? Please also cite this study in the paper under Discussion.

Liu P, Xu M, Cao L, Su L, Lu L, Dong N, Jia R, Zhu X, Xu J. Impact of COVID-19 pandemic on the prevalence of respiratory viruses in children with lower respiratory tract infections in China. Virology Journal. 2021 Dec;18(1):1-7.

  1. Author has used different nucleic acid extraction kits for isolating SARS-CoV-2 and other respiratory viruses. What is the purpose of using different kits while the sample would be the same for detecting all the viruses. It may indirectly affect the result. Justify.

  1. What does it mean by non-pharmaceutical interventions (Discussion) ? Please elaborate.

  1. Table 2, Please mention the total number of positive cases in column 2 (Jan. 2017-Feb. 2020) and column 3 (Mar. 2020-May. 2021). It is hard to see how the percent positivity for individual viruses is calculated.

  1. There are grammatical errors and incomplete sentences in the text. Please carefully go through the text and correct these errors. For examples

Page 2, lines 45-47 represent an incomplete sentence as “Furthermore, to contain COVID-19 infections, the Italian government …..cases numbers.  If I understand correctly, the Italian government derived a different colour based system to differentiate COVID-19 cases for proper reporting of COVID-19 patients. Please rewrite the sentence in a way that makes the right sense.

Page 5, lines 120-123 please correct sentence “Group 45-64 years was more prevalent during pandemic period compared……… It should be “respiratory viruses were more prevalent in the age group of 45-64 years patients.

Round 2

Reviewer 3 Report

Author has incorporated suggested changes. Article seems suitable for consideration.